# Interoception in Old Age

**DOI:** 10.3390/brainsci12101398

**Published:** 2022-10-17

**Authors:** Gili Ulus, Daniela Aisenberg-Shafran

**Affiliations:** 1Department of Clinical Psychology of Adulthood and Aging, Ruppin Academic Center, Emek Hefer 4025000, Israel; 2The Dror (Imri) Aloni Center for Health Informatics, Ruppin Academic Center, Emek Hefer 4025000, Israel

**Keywords:** old adults, interoception, emotional experience, physiological arousal

## Abstract

Emotion regulation in old age was found to be more efficient; seniors seem to focus less on the negative aspects of experiences. Here, we ask, do older individuals regulate their emotions more efficiently or are they numb to the physiological changes that modulate these emotions? Interoception, the perception of physical feelings, influences a person’s mood, emotions, and sense of well-being, and was hardly tested among older adults. We examined the awareness of physiological changes (physiological arousal—blood pressure and heart rate) of 47 older adults, compared to 18 young adults, and their subjective reports of emotional experiences while viewing emotional stimuli. Interoception was decreased in old age. Blood pressure medications had a partial role in this reduction. Moreover, interoception mediated emotional experience, such that low interoception led to lower experiences of changes in physiological arousal. These findings may account for the emotional changes in old age, suggesting a decline in sensitivity with age, which leads to a positive interpretation of information.

## 1. Introduction

Emotional reactivity when encountering emotional stimulation involves changes in the experience, the activity of the autonomic nervous system (physiological arousal), and behavior. For example, an emotional response to a threatening situation will result in a feeling of fear, a rapid heartbeat, and a frightened expression [1].

In the nineteenth century, William James [2] suggested a close relationship between one’s self-perception of physiological arousal and the resulting emotional experience, assuming the former to be significant to the latter. Other studies on emotion science throughout the last century argued differently, i.e., representations of bodily changes were outcomes rather than precursors to emotions [3,4,5,6]. Growing evidence suggests that the perception of change in the physiological arousal occurring in the body, termed interoception, may be a significant component of emotion [7,8,9,10,11,12,13]. However, relevant theories on bodily feedback remain controversial [14].

### 1.1. Interoception

Interoception is the perception of physical sensations (such as heartbeat, breathing, stomach sensations, and autonomic nervous system activity) that affect an individual’s mood, psychological well-being, and feelings [15]. In recent years, interoception was separated into various aspects [16], such as interoceptive awareness and interoceptive accuracy [16,17,18,19]. Interoceptive awareness reflects the perception of physiological changes in the body [17] and is measured subjectively using questionnaires [20] that require individuals to report if they feel physiological changes in response to different situations [21,22,23]. Interoceptive accuracy reflects the ability to identify the changes in physical functions accurately using objective measures, such as the heartbeat perception task [20]. The heartbeat perception task [20], requires individuals to report the number of times they identify their heartbeat within a limited time frame, via the heartbeat tracking task [24]. Another way is to report when they feel their heartbeat through perceived synchrony of the heartbeat with external stimuli (e.g., tone), via the heartbeat discrimination task [25,26]. These measurements were found to involve the anterior insula, which is related to interoception ability [20,27,28,29,30].

The association between interoception ability (awareness and accuracy) and emotional experience is well established in studies that measure brain activity [1,9,31,32,33]. In addition, it was found that higher interoception is associated with negative emotions [27], alexithymia [34], higher levels of emotional arousal [28,29,35,36,37], better self-regulation capacities [38], and better emotion regulation in response to negative effects [39]. Together, these findings support the suggestion that the detection of bodily sensations can affect emotional experiences [15]. However, physiological changes, emotional experiences, and interoception ability were found to behave differently among older adults. Hence, it would be interesting to understand their relations in the old population.

The main objective of the next section is to address changes in interoception ability and emotional experiences.

### 1.2. Aging and Physiological Arousal

Aging is associated with a progressive decline in various physiological processes [40]. The autonomic nervous system (ANS), which plays an important role in emotion, is clearly affected by age [41,42]. For example, autonomic reactivity decreases with age [40] due to changes in blood circulation [43].

Studies that examined the physiological response of the ANS while individuals viewed emotionally salient pictures have found that the reactivity changes as a function of the valence and arousal levels of the pictures (e.g., [44,45]). Heart rate (HR) deceleration appears in the first few seconds of the picture display [45,46]. HR deceleration is generally greater when viewing negative pictures (compared to positive or neutral pictures) [44,45,47,48,49]. There are, however, inconsistent findings regarding changes in blood pressure (BP) when viewing emotionally salient pictures. In other words, studies have shown an increase in BP as a result of arousal levels only [48], as a result of arousal and valence levels [50], and as a result of valence only [49].

Older individuals have been found to exhibit decreased (e.g., [47,51]), increased (e.g., [51,52,53,54]), and similar (e.g., [41]) physiological reactivity to emotional stimulation when compared to younger adults.

### 1.3. Aging and the Emotional Experience

Studies have shown changes in emotional reactivity with age, such as more effective emotion regulation [55,56], reduced focus on negative emotions [57,58], and a higher motivation to maintain positive affectivity [59]. However, some studies that used emotionally salient pictures to examine the differences in the emotional experiences between young and older adults have found inconsistent results. In some studies, older adults rated the pictures more positively than younger adults [60]. In others, older adults reported being more aroused by the pictures than younger adults [51]. While in others, older adults rated the pictures less positively and reported being less aroused than younger adults [61]. In general, participants reported higher arousal in response to negative pictures than to positive pictures and reported more pleasant emotional experiences in response to positive pictures than to negative pictures [44,46,51,62,63].

Inconsistent findings were also found in self-reported emotional experiences and in the physiological arousal levels across different age groups [44,51,64,65,66,67,68].

Various theories have attempted to settle the above inconsistencies by considering emotion regulation [69,70,71,72]. For example, the socioemotional selectivity theory (SST), a lifespan theory of motivation [70], suggests that goals change as a function of future time horizons. When people perceive their time left to live as lengthy, as is usually the case for young adults, information goals are most important in preparing for the range of uncertain challenges that the future holds. When the time left to live is perceived as shorter, as is usually the case for older adults, emotional goals (i.e., emotional meaning, emotion regulation, and psychological well-being) assume primacy. This change of focus leads to experiences of more positive emotions and less negative emotions over time [73]. This does not mean that emotion regulation is characterized by hedonism, but rather by a complex mix of positive and negative emotions [74]. For more support for the positive effects in old age, see the Selective Optimization with Compensation theory—SOC; [69]. Further, Charles [71] suggests that older adults generally respond better than younger adults to emotional stimuli because they regulate their emotions by avoidance or via reduced exposure to emotional distress. She adds that when emotional stimuli are very arousing, this positive effect decreases for older adults, as it is difficult to regulate emotions in such highly arousing circumstances. It is important to note that not all findings or inconsistencies in the findings can be understood through current theories on emotional motivation and regulation strategies.

From a different point of view, Mendes [75] suggests maturational dualism. This idea suggests that a weakened connection between the body and mind is a major factor affecting emotional experiences in old age. According to Mendes, interoceptive ability in old age decreases because the peripheral nervous system becomes more vulnerable, and physiological reactivity becomes blunted. Therefore, older adults are less sensitive to perceptions of physiological changes triggered when emotionally stimulated. When it is not possible to identify internal physiological changes, reactivity to emotional stimulation is solely reliant on the emotional value of the stimulus and external cues from the environment. Mendes suggests that the tendency of older individuals to attend to positive stimuli and to avoid negative stimuli, as the socioemotional selectivity theory (SST) [70] suggests, might be partly motivated by an inability to perceive internal physiological information. Mendes’ idea of maturational dualism proposes a possible explanation for the gap between physiological arousal and emotional experiences in old age by suggesting a mediating role of interoception. So far, only one experimental study [72] has been conducted to examine this relationship in its entirety, as we will elaborate on now.

### 1.4. Aging and Interoception

Several studies have evaluated interoception in old age. Some found increased awareness of the feeling of stomach fullness [76], pain [77,78], and touch [79]. Mixed results were found regarding the level of awareness of rectum activity [80,81]. Other studies found that interoceptive accuracy decreases in old age, as reflected in a significant decline in the ability to identify heart rate [82,83]. To the best of our knowledge, only a few studies in recent years have examined interoception in old age [72,84,85,86]. Low performances in the heartbeat detection task (interoceptive accuracy) were found for older adults [85], though the small sample of old adult participants (6 participants aged 60–63 out of 59 participants aged 22–63) made the statistical power of this study questionable [72]. In another study, Lohani [72] examined the influence of interoceptive ability on the association between emotional experience and physiological arousal in aging by using sadness manipulation. She found no association between interoceptive ability and the association between physiological reactivity and emotional experiences in old age. Interoceptive accuracy was measured via the heartbeat detection task to compare young adults (aged 18–23) and older adults (aged 60–87) in their interoceptive abilities. The author found that the age variable accounted for only 5% of the variance in performance of the task, which limits any interpretations that can be made about older adults’ interoceptive abilities and the association between emotional experience and physiological arousal. Nevertheless, Lohani [72] concluded that the results could reflect a true decline in interoceptive ability or difficulty for older adults to perform the heartbeat detection task. Two other studies [84,86] found that interoceptive awareness, measured by the body perception questionnaire (BPQ; [22]), declines with age. Murphy et al. [86] also examined interoceptive accuracy by the heartbeat detection task and found that the variance in the performance of the task, as explained by age, was modest. This considered, they still concluded that interoceptive accuracy decreases with age. Taken together, it is unclear whether the heartbeat perception task is suitable for testing interoceptive ability in old age.

### 1.5. The Present Study

The accurate measurement of *interoceptive accuracy* in old age remains an unsolved problem. Therefore, the present study examined the version of *interoceptive accuracy* through a combination of subjective reports of physiological arousal and the objective measurement of the physiological arousal (i.e., blood pressure (BP) and heart rate (HR)), generated while participants viewed emotionally salient pictures. Each set of pictures was displayed for 30 s, 5 s per picture, and participants were asked to rate the change they noticed in physiological arousal at the end of each set (every 30 s). In addition, actual measures of physiological arousal were assessed, measuring BP change and HR change (more information about the emotional stimulation will be provided in the method section of Experiment 1).

To the best of our knowledge, no existing study has examined interoception via manipulation of emotional stimuli. Dunn et al.’s [36] and Pollatos et al.’s [29] studies on young adults demonstrated that a high level of interoception is associated with a positive correlation between heart rate reactivity and subjective emotional arousal ratings of emotional pictures. However, these researchers did not use emotional stimuli as a means to examine interoception. In our research, we also used the body perception questionnaire short form (BPQSF) [22,84], with an emphasis on the body awareness (BA) sub-scale. This instrument offers a subjective report of general awareness of internal physiological changes (i.e., interoceptive awareness), and has been used in various studies to examine interoceptive ability [16,27,34,84,86,87,88,89,90].

The general aim of this study was to examine whether interoceptive ability, in general, could explain the body–mind gap in old age, which is reflected in the consistent inconsistency in findings regarding the association between physiological changes and emotional experience. More specifically, our aim for Experiment 1 was to examine if there is a decrease in interoception ability in old age, while our aim for Experiment 2 was to examine if a change in interoception affects the emotional experience.

Even though emotion regulation is considered better in older adults, and is associated with interoception, there is no sufficient evidence in the literature for this association in older adults. Mendes (2010) has suggested that interoception decreases in old age, which raises the question—what is the true basis of the ‘improved’ emotion regulation in older age, and is it indeed a better response or lack of awareness that does not evoke the negative emotion?

Our research hypotheses were: (Experiment 1) Interoception ability is affected by aging, such that: (a) the changes in self-reported physiological arousal after 30 s and then every 30 s, and the changes in physiological arousal (i.e., BP and HR) at baseline (relative to every 30 s during the task), would be positively associated in the young adults’ group, but not in the older adults’ group (i.e., interoceptive accuracy) (in line with [36]); (b) older adults would report lower levels of body awareness measured by the BA sub-scale (i.e., interoceptive awareness) compared to young adults (in line with [84,86]). Further, and in accordance with Mendes’ idea of maturational dualism [75], (Experiment 2), we expect that (c) in older adults, a lower level of interoceptive awareness, at baseline, would be associated with greater levels of pleasant experiences during the emotional stimuli task overall, and with (d) fewer changes in physiological arousal measured by BP and HR during the emotional stimuli overall.

The role of interoception is presented in Figure 1, showing the possible contradicting outcomes of the theories discussed above.

In addition, the experimental method we chose for emotional stimulation raises secondary hypotheses (e.g., [46]) and so we hypothesized that (e) the reporting of arousal in response to negative pictures will be higher than in response to positive pictures (examined in Experiments 1 and 2); (f) the reporting of a pleasant emotional experience in response to positive pictures will be higher than in response to negative pictures (examined in Experiment 2); and (g) a more prominent deceleration in HR will occur when viewing negative pictures compared to positive pictures (as examined in Experiments 1 and 2).

## 2. Materials and Methods- Experiment 1

This experiment examined whether there was a decrease in interoception ability in old age.

### 2.1. Method

A mixed design was developed with one between-group independent variable: age (young adults vs. older adults), and two within-group independent variables: picture type (negative vs. positive) and picture content (14 picture sets). Two dependent variables were examined: (1) interoception accuracy was measured by the association between two indicators: objective measures of physiological arousal (i.e., BP and HR) and subjective self-report of changes or activity in physiological arousal. (2) Interoceptive awareness was measured by the BA sub-scale (BPQSF; [22,84]).

### 2.2. Participants

A total of 38 participants, 18 young adults (age range 20–34, M_age_ = 25.72, SD_age_ = 3.21, 13 females) and 20 older adults (age range 73–87, M_age_ = 79.65, SD_age_ = 5.72, 10 females), took part in the experiment. Power analysis referred to F tests of repeated measures, within–between interaction, with an estimated medium effect size of 0.25, which yielded a total of 34 participants, for both age groups. The older adult participants were recruited from day centers and kibbutzs in the Gaza section of Israel. The young adult participants were first-year BA students recruited at the Ruppin Academic Center as part of their course credit in “Psychology Introduction”.

### 2.3. Procedure

The experiment was administered in a quiet room and for each participant separately. Upon arrival, participants provided informed consent and were debriefed upon task completion. Information regarding their physiological measures was collected. The older participants were administered the Mini-Mental State Examination (MMSE) [91] in Hebrew [92], to assess their cognitive level, and all scored 25/30 points and above. All participants completed a short demographic and medical questionnaire. Then, the interoception questionnaire and the interoceptive task were completed. To avoid order effects, half of the participants in each age group completed the questionnaire prior to the task, and half completed the questionnaire at the culmination of the task. The blood pressure (BP)–heart rate (HR) watch was worn on the non-dominant hand of each participant and physiological arousal at rest was measured for 3 min. Next, the participants were told that for 20 min they would watch a series of pictures and were asked to rate the activity or changes in their physiological arousal on the scales that appear after each set of pictures. A training set was then administered, including 6 positive pictures (from one of the positive sets), followed by 14 experimental sets. Each set of pictures appeared twice in a random order, with the restriction that no more than two sets of the same emotional valence were presented consecutively (see the Pearson correlations between the measurements tested during the task in Table A1 in the Appendix B). Between the experimental sets, there was a pause of 2 s in which a dot mark appeared on the screen. After the pause, the physiological arousal rating scale appeared. In the end, the researcher thanked the participants for their cooperation.

### 2.4. Measures

#### 2.4.1. Emotional Stimuli

The emotional stimuli used in this study were largely based on the emotional stimuli used in Gomez et al.’s [93] study, which examined young and old participants, and were taken from the International Affective Picture System (IAPS) [94] and the Nencki Affective Picture System (NAPS) [95,96,97,98] databases. Participants viewed 84 pictures, and each set of pictures was presented twice, resulting in a total of 168 pictures. The dimensions of the images were 1024 × 768 mm and 1600 × 1200 mm. All pictures were of good quality and were presented in color. Further, each set of images included an equal number of objects in the pictures. The pictures were divided into 14 sets, each set containing 6 emotionally salient pictures. A total of 7 sets of pictures contained positive emotional valence (i.e., pleasant; arousal level: M = 4.69, SD = 2.11, emotional valence: M = 7.25, SD = 1.46) and 7 sets of pictures contained negative emotional valence (i.e., unpleasant; arousal level: M = 5.91, SD = 1.70, emotional valence: M = 2.89, SD = 1.31), according to the norms of IAPS [94] and NAPS [95]. The 7 positive sets included content such as stimulating desserts, heterosexual couple love, babies, happy children, pleasant nature, pets, and sports scenes. The 7 negative sets included content such as environmental pollution, corruption, burned bodies, suffering, dead animals, unpleasant family scenes, violence, negative emotion, negative situations, and older adults, which received negative valence according to the IAPS norms.

Each set of pictures was displayed for 30 s, 5 s per picture. The six pictures within each set appeared in random order. The experiments were built using the software OpenSesame [99].

#### 2.4.2. Physiological Index

The BP and HR indices were recorded during the 3 min of baseline before the task began and during the pictures-viewing task every 30 s. The BP and HR watches (FITNESS TRACKER) were measured by monitoring equipment (“WearHealth” software). The systolic (BPSYS) and diastolic (BPDIA) BP and HR were calculated separately at each measurement time (baseline and in the course of the task), and a joint average of the differences of each measurement time was calculated to assess the change of arousal at each measurement [100,101].

### 2.5. Interoception Measures

#### 2.5.1. Self-Report

Interoceptive accuracy was measured by the rating of identifying internal physiological activity or changes during the emotional task compared to actual physiological arousal that was measured after each set of pictures. The actual arousal change measure was conducted using the body-attached watch that measured BP and HR. The self-reports of arousal were compared to the watch data.

Participants rated their physiological experience based on the statement, “Mark the level at which you are now experiencing activity or changes in the physical arousal that occurs within you.” The rating scale consisted of 9 points, ranging from 0 (no activity or changes in arousal at all) to 8 (large activity or changes in arousal).

#### 2.5.2. Body Perception

The body perception questionnaire short form (BPQSF) [22,84] was used to measure the participants’ awareness of internal physiological changes in most situations (interoceptive awareness). This questionnaire consists of 46 statements. A total of 26 statements reflect body awareness (BA), a measure of awareness of physiological bodily processes. A total of 14 statements reflect a supradiaphragmatic reactivity (SUPRAR), a measure of the stress reactivity of autonomically-innervated organs above the diaphragm. The responses to these 14 statements reflect a regression of neural circuits that promote social involvement and restful states of relaxation as well as a “fight or flight” response in the sympathetic system. Six other statements reflect the perception of bowel activity of subdiaphragmatic reactivity (SUBR) [22]. Both SUPRAR and SUBR sub-scales provide information about bodily stress reactivity [84]. Our focus was on the BA sub-scale as an indicator of interoception ability [84]. An example of a question is: “During most situations I am aware of: How fast I am breathing; stomach and intestines, etc.”. Responses were recorded on a 5-point Likert scale based on the frequency (“never” to “always”) with which the subject perceived this awareness. “Always” indicates a high awareness of changes in the body while “never” indicates a low awareness of changes in the body. Perception of the body was calculated by the sum of the ratings of each participant, such that a higher score suggests higher interoceptive ability.

Internal consistency reliability was taken from Cabrera et al. [84] and was assessed from the data on the American college sample (N = 315, BA: categorical ω = 0.92, SUPRAR: categorical ω = 0.88, SUBR: categorical ω = 0.78). Before the experiment began, the BPQSF questionnaire was administered as a pilot on a psychology student sample at the Ruppin Academic Center (N = 30, M_age_ = 29.4, SD = 5.76, range = 24–53, 20 female). This pilot was used for reliability testing (BA: a = 0.90, SUPRAR: a = 0.81, SUBR: a = 0.76). In the current study the internal consistency reliability of Cronbach’s alpha was BA: a = 0.94, SUPRAR: a = 0.78, SUBR: a = 0.81.

In the first experiment: Body awareness (BA) sub-scale scores ranged from 36 to 120 (M = 70.54, SD = 20.15), supradiaphragmatic reactivity (SUPRAR) sub-scale scores ranged from 15 to 40 (M = 24.9, SD = 6.3) and Subdiaphragmatic reactivity (SUBR) sub-scale scores ranged from 6 to 21 (M = 11.11, SD = 3.87). In the second experiment: BA sub-scale scores ranged from 32 to 114 (M = 55.78, SD = 17.17), SUPRAR sub-scale scores ranged from 15 to 37 (M = 20.74, SD = 6.55), and SUBR sub-scale scores ranged from 6 to 20 (M = 9.85, SD = 3.57).

#### 2.5.3. Demographics and Health Questionnaire

Demographic data, such as age (years), gender (male, female), weight, family status, education (years), living area, work or volunteer status, smoking status, and daily alcohol consumption, were collected. The health questions were related to alcohol consumption, daily cigarette smoking, and the use (and duration of use) of BP medications and psychiatric medications.

### 2.6. Statistical Analyses

Our analyses involved repeated measure univariate analyses of variance (ANOVAs) for the physiological arousal and self-reported arousal dependent variables. Independent variables were age (young/old), a between-participant variable and picture type (negative/positive), or picture content (14 contents), as within-participant variables. We also conducted Pearson correlations (two-tailed) to examine the association between physiological arousal index, self-reported arousal, and the BPQSF sub-scales. Simple linear regression analyses were used to examine the explained variance of the correlation findings. Other analyses of variance and correlations were used to examine demographic variables with the interoception measures. The statistical tests were conducted using SPSS version 25 (IBM Corp, Armonk, NY, USA).

### 2.7. Results

Mean physiological measurements were calculated for each participant, as well as the sum of self-reports for interoception. One young participant was removed from the data analysis due to extreme results (more than 3 SD from the mean). All data can be viewed in the Appendix A.

#### 2.7.1. Demographic Variables

Demographic variables were tested (see Table 1). A significant effect was found for BA sub-scale and gender (*t* (35) = −2.65, *p* < 0.05, 95% CI [−29.56,−3.89]), which suggests that females were more aware of their bodies (M = 76.87, SD = 20.98) than were males (M = 60.14, SD = 13.85). Another significant negative correlation was found between weight and the two sub-scales, BA (r = −0.35) (F _(1,35)_ = 4.94, *p* < 0.05; Adj. R^2^ = 0.09; β = −0.35) and SUPRAR (r = −0.35) (F _(1,35)_ = 4.75, *p* < 0.05; Adj. R^2^ = 0.09; β = −0.35), such that higher weight was associated with a decrease in body awareness and with decreased autonomic nervous system reactivity above the diaphragm (i.e., stress responses). We found that BP medication use was significantly different for all three physiological indices: HR change, BPDIA change, and BPSYS change (see Table 2). These results indicate that participants who used BP medication showed fewer changes from baseline in all three physiological indices compared to participants who did not use BP medication. No other demographic effects were found.

#### 2.7.2. Testing Hypothesis a: Interoceptive Accuracy Compared between Young and Older Adults Using Physiological Arousal and Self-Report

Consistent with our first hypothesis, significant correlations were found between the three physiological indices: HR change, BPDIA change, BPSYS change and self-reported physiological arousal for the young adults only (see Table 3). Three simple linear regression analyses showed that 21% of the self-reported physiological arousal was explained by HR (F _(1,15)_ = 5.33, *p* < 0.05; Adj. R^2^ = 0.21; β = 0.51), 33% was explained by BPDIA (F _(1,15)_ = 8.76, *p* < 0.05; Adj. R^2^ = 0.33; β = 0.61) and 23% was explained by BPSYS (F _(1,15)_ = 5.84, *p* < 0.05; Adj. R^2^ = 0.23; β = 0.53). These findings indicate that less marked HR and BP deceleration was associated with self-reported higher physiological arousal among young adults, in line with Dunn et al. [36]. No significant correlation was found for the older group, yet a negative directional trend was observed, opposite to the findings in young adults (see Table 3 and Table 4).

#### 2.7.3. Testing Hypothesis b: Interoceptive Awareness Compared between Young and Older Adults

The lower part of Table 1 describes the results of the independent t-test (two-tailed) conducted for the BPQSF subjective ratings by the two age groups. Overall, the t-test indicated a significant effect of the age group on the subjective rating of the BA sub-scale, with younger participants indicating greater awareness of their bodies than older participants. This result is consistent with our hypothesis that interoception decreases with age. No significant effects were found between the BA sub-scale and the physiological indices, or with self-reported physiological arousal. This is in line with the understanding that there are several types of interoception.

Interestingly, we found a significant effect for the BA sub-scale and BP medications use (F _(1,35)_ = 21.37, *p* < 0.001, ηP^2^ = 0.38), suggests that participants who use BP medications (M = 51.73, SD = 9.19) were less aware of their bodies than participants who do not use BP medications (M = 78.50, SD = 18.44).

In a post hoc analysis, we examined whether there is a difference in body awareness between three groups: (1) Young adults (no usage of BP medications; *n* = 17); (2) older adults (no usage of BP medications; *n* = 9); and (3) older adults (with the usage of BP medications; *n* = 11). We found a significant difference between the groups (F _(2,34)_ = 13.65, *p* < 0.001, ηP^2^= 0.45). Specifically, young adults reported the highest level of body awareness (M = 82.94, SD = 17.65), older adults with no usage of BP medications reported less body awareness (M = 70.11, SD = 16.86), and older adults with the usage of BP medications reported the lowest level of body awareness of all groups (M = 51.73, SD = 9.19). Multiple comparisons analysis showed significant differences between older adults (with the usage of BP medications) and the two other groups (young; *p* < 0.001, old; *p* < 0.05), with no significant difference between the older adults (no usage of BP medications) and the young adults in their body awareness (*p* = 0.15).

The results raise a question about the role of BP medications on interoception ability, hence we decided to examine whether the relationship between interoception and age was mediated by BP medication use. First, we examined the correlations between these tree variables, which were significant (see Table 5). An exploratory mediation model was tested using a series of linear regression analyses. The results of this model are illustrated in Figure 2. As can be seen, there was a significant effect of age on the body awareness (BA) sub-scale (F _(1,35)_ = 19.74, *p* < 0.001; Adj. R^2^ = 0.34; βc = −0.60) and on BP medication use (F _(1,35)_ = 17.13, *p* < 0.001; Adj. R^2^ = 0.31; βa = 0.57). Once BP medication use was inserted as another predictor, the entire model was found to be significant (F _(2,34)_ = 15.10, *p* < 0.001; Adj. R^2^ = 0.44; βb = 0.404; βc’ = −0.37), suggesting partial mediation. A Sobel test demonstrated significant differences between the predictors (Sobel’s statistic = −2.34, *p* < 0.05). The model explains 44% of the variance in body awareness. Though not part of our hypotheses, these results indicate the significant effect of the use of BP medications on interoceptive awareness in old age.

#### 2.7.4. Secondary Hypothesis e: Arousal Reports in Response to Negative Pictures Were Higher Than in Response to Positive Pictures

Table 6 shows the means and standard deviations for the self-report physiological arousal ratings of the pictures for the two age groups. We found a significant main effect of picture type (F _(1,35)_ = 35.44, *p* < 0.05; ηP^2^ = 0.50), which suggests that negative pictures were reported as more physiologically arousing than positive pictures by all participants, in line with our hypothesis and evidence from the current literature [44,51,63]. The main effect of age group was also significant (F _(1,35)_ = 5.12, *p* < 0.05; ηp^2^ = 0.13), suggesting that older adults reported higher arousal compared to young adults, in accordance with previous findings [51]. We also found a significant main effect of picture content (F _(13,455)_ = 19.82, *p*< 0.001; ηp^2^ = 0.36). Post hoc analyses using Bonferroni revealed that different contents encouraged different levels of self-reporting.

No interaction between age group and picture type was found in the self-report of physiological arousal. However, a significant interaction was found between picture content and age group (F _(13,455)_ = 3.23, *p* < 0.001; ηp^2^ = 0.08). Post hoc analyses (Bonferroni) revealed different levels of physiological arousal of subjective reports between the age groups in response to picture content. That is, old people reported a greater change in physiological arousal compared to young people (see Table A3 in Appendix B). The contents that showed age differences were both positive and negative.

#### 2.7.5. Secondary Hypothesis g: HR Prominent Deceleration Occurred When Viewing Negative Compared to Positive Pictures

No main effects or interactions were found in the physiological index for the two age groups (for picture type or content). However, we did find an insignificant trend of deceleration in HR change for negative pictures compared to positive pictures, in line with our secondary hypotheses and the literature findings (e.g., [36,47]). This trend in deceleration was only observed in the group of young adults. In addition, it seems that young adults showed a more prominent deceleration (non-significant) in all physiological measures as opposed to older adults (see Figure A1 in Appendix B). Further, the group of older adults who did not use BP medications showed similar physiological arousal levels to those of young adults (without BP medications), (see Figure A2 in Appendix B).

#### 2.7.6. Interim Discussion

The purpose of Experiment 1 was to examine whether interoception ability decreases with age. The measure of interoceptive accuracy that examined the correlation between self-reported physiological arousal and objective physiological arousal was significantly positive for young adults, but only demonstrated a negative directional trend, an insignificant correlation, for older adults. This finding is in line with several studies [29,36], which found a similar positive association between the subjective reporting of arousal and objective physiological arousal among participants with high interoception, and a negative association of these variables with low interoception ability. It is possible that the lack of statistical significance of the negative correlation between physiological arousal and self-reporting of physiological arousal was due to the large intra-individual variance observed in older adults [102,103]. Importantly, another possibility is that the lack of correlation in the old group is an indication of a decline in interoception. However, the measure of interoceptive awareness, as measured by the BPQSF, displayed a significant difference between the age groups, which suggests that interoceptive awareness decreases in old age, as indicated in the literature [84,86]. In brief, while the results support a reduction in interoceptive awareness in old age, the correlation between subjective and objective physiological arousal (interoceptive accuracy) is still worthy of further investigation.

We also found that the BP medications use played a significant role in interoceptive awareness, as there was a significant difference between participants who used BP medications and those who did not use BP medications in the BA sub-scale. Interestingly, those who did use BP medication reported higher interoception with no significant difference between the two age groups. A partial mediation model was revealed, such that BP medication use mediated the association between age and interoception and explains 44% of the variation. This finding may support a blunting effect of BP medication use on internal physiological processes and interoception ability.

Our results seem to support the idea by Mendes [75], i.e., the peripheral nervous system becomes more vulnerable with age. The uses of BP medications seem to affect the cardiovascular system [104]. This may encourage less sensitivity to the physiological changes triggered as a result of emotional stimulation and may require the individual to rely on external cues from the environment to report physiological arousal [75]. This explanation may also shed light on why older adults report higher levels of arousal than do young adults, due to external cues.

Consistent with our secondary hypotheses, higher arousal was reported in response to negative pictures when compared to positive pictures, in line with previous findings (e.g., [36,66]). Though non-significant, an HR deceleration trend was observed when viewing the pictures, consistent with an initial orienting response (e.g., [45]). Only within the young group was HR deceleration greater (but not significantly different) for negative pictures than for positive pictures. When the BP medication use variable was held constant, we observed a similar HR decrease for negative pictures compared to positive pictures in older adults. This trend of HR deceleration is well established in the literature [29,45,46,105].

## 3. Materials and Methods- Experiment 2

Based on the findings of the first experiment, which showed a decrease in interoception ability in old age (interoceptive awareness), this experiment will examine if these changes affect the emotional experiences of older adults, affecting the association between physiological arousal and emotional experience. We expected that lower interoception ability would be accompanied by better emotional experiences and fewer changes in physiological arousal (in accordance with Mendes [75]).

### 3.1. Method

A within-group design was developed with three independent variables: picture type (negative vs. positive), picture content (14 picture sets), and interoception (measured by the BPQS; [22,84]). Two dependent variables: (1) emotional experience (measured by the subjective self-report regarding valence (between very pleasant and unpleasant) and arousal (between very aroused and very calm) [94], and (2) physiological arousal (measured by BP and HR).

### 3.2. Participants

A total of 27 old adult participants, who did not participate in Experiment 1, took part in this experiment (age range 70–86 years, M_age_ = 77.81, SD_age_ = 5.7, 15 females). Participants were recruited from day centers and kibbutzim in the Gaza section of Israel.

### 3.3. Procedure

The procedure was the same as in Experiment 1, except for the self-report questionnaire. Here, rather than reporting changes in physiological arousal as part of the interoceptive accuracy, participants reported emotional experience and emotional arousal following viewing the picture sets.

### 3.4. Emotional Stimuli

Similar to Experiment 1.

### 3.5. Measures

Physiological index, BPQSF (with internal consistency reliability of Cronbach’s alpha; BA: a = 0.90, SUPRAR: a = 0.85, SUBR: a = 0.66), and the Demographics and Health Questionnaire were collected in a similar manner to Experiment 1.

### 3.6. Emotionally Subjective Reports Index

In order to measure the participant’s subjective emotional experience, we used the emotional valence and emotional arousal scales from self-assessment (Manikin, Sam, [94]). The emotional valence scale ranged from 0 (not pleasant at all) to 8 (very pleasant) and the emotional arousal scale ranged from 0 (very calm) to 8 (very arousing).

### 3.7. Statistical Analysis

Our analyses were similar to Experiment 1, with two different dependent variables: emotional experience and emotional arousal (instead of interoceptive accuracy within the task). Physiological arousal remained a dependent variable.

### 3.8. Results

Mean physiological measurements were calculated for each participant, as well as the sum of self-reports for interoception (interoceptive awareness), emotional arousal, and emotional experience.

#### 3.8.1. Demographic Variables

Demographic variables were again tested (see Table 7). Similar to Experiment 1, a significant effect was found for the BA sub-scale and gender. Significant effects for SUPRAR and SUBR sub-scales were also found (see Table 8). Female participants reported higher body awareness and greater ANS reactivity above and below the diaphragm. A marginally significant interaction effect was found between gender and picture type for emotional experience (F _(1,25)_ = 3.80, *p* = 0.067; ηP^2^ = 0.13). This suggests that females had a higher unpleasant emotional experience (M = 0.67, SD = 0.98) than males when viewing negative pictures (M = 1.58, SD = 1.50; *p* = 0.067); and similar pleasant emotional experiences (M = 6.42, SD = 0.79) when viewing positive pictures as men (M = 6.67, SD = 0.90; *p* = 0.45).

A significant negative correlation was found between weight and the SUBR sub-scale (r = −0.39) (F _(1,25)_ = 4.50, *p* < 0.05; Adj. R^2^ = 0.12; β = −0.39), showing that a higher weight was correlated with reports of lower ANS reactivity below the diaphragm. Unlike Experiment 1, no significant correlation between BP medication uses and the BA sub-scale was found. No other demographic effects were found.

#### 3.8.2. Testing Hypothesis c: Interoceptive Awareness and Pleasant Experiences during the Emotional Stimuli Task

Consistent with our main hypothesis, a significant negative correlation was found between self-reported emotional experience and the BA sub-scale (r = −0.43) (F _(1,25)_ = 5.69, *p* < 0.05; Adj. R^2^ = 0.15; β = −0.43), indicating that when participants reported lower awareness of the body before the task, they reported more pleasant emotional experience during the task.

Looking only at negative pictures, a significant negative correlation was found (r = −0.38) (F _(1,25)_ = 4.32, *p* < 0.05; Adj. R^2^ = 0.11; b = −0.38), such that individuals who reported lower body awareness also reported less unpleasant emotional experience. In other words, we found evidence for a specific directional association between interoception and emotional experiences in old age, such that low interoception is associated with less negative emotional experiences.

#### 3.8.3. Testing Hypothesis d: Association between Interoceptive Awareness and Changes in Physiological Arousal during the Emotional Stimuli

Another significant negative relationship was found between the BA sub-scale and mean change in BPDIA (r = −0.38) (F _(1,25)_ = 4.25, *p* < 0.05; Adj. R^2^ = 0.11; β = −0.38), as well as with mean change in BPSYS (r = −0.38) (F _(1,25)_ = 4.24, *p* < 0.05; Adj. R^2^ = 0.11; β = −0.38). In addition, a marginally significant correlation was found between mean change in HR and BA sub-scale (r = −0.37, *p* = 0.058) (F _(1,25)_ = 3.96, *p* = 0.058; Adj. R^2^ = 0.10; β = −0.37). This means that reports of low body awareness corresponded with smaller changes in physiological arousal from baseline, suggesting that low interoception was associated with fewer changes in physiological arousal.

#### 3.8.4. Secondary Hypothesis e and f: Arousal Reports to Negative Pictures Were Higher Than in Response to Positive Pictures, and Reports of Pleasant Emotional Experiences in Response to Positive Pictures Were Higher Than in Response to Negative Pictures

Means and standard deviations were calculated for the subjective ratings of emotional experience and emotional arousal, in response to picture type and picture content (see Table 9 and Table 10). As we expected, significant main effects of picture type for emotional experience (F _(1,26)_ = 306.85, *p* < 0.001; ηp^2^ = 0.92) and for emotional arousal (F _(1,26)_ = 16.25, *p* < 0.001; ηp^2^ = 0.38) were found. That is, negative pictures were reported as less pleasant and more arousing than positive pictures. We also found significant main effects of picture content for emotional experience (F _(13,338)_ = 133.30, *p* < 0.001; ηp^2^ = 0.84) and for emotional arousal (F _(13,338)_ = 12.24, *p* < 0.001; ηp^2^ = 0.32), revealing (using post hoc Bonferroni analyses) different levels of subjective reports in response to picture content.

#### 3.8.5. Secondary Hypothesis g: HR Deceleration While Viewing Negative Compared to Positive Pictures

No significant differences in physiological arousal were found for the picture type or content. No significant correlations were found between emotional experience and physiological measures; hence, it would be interesting to see whether interoception may explain this gap. Two significant negative correlations were found. First, between the emotional arousal reports and the HR change (r = −0.42) (F _(1,25)_ = 5.47, *p* < 0.05; Adj. R^2^ = 0.15; β = −0.42). Second, between the emotional arousal reports and the mean change in BPSYS (r = −0.43) (F _(1,25)_ = 5.62, *p* < 0.05; Adj. R^2^ = 0.15; β = −.43). These correlations suggest that reports of high emotional arousal corresponded with a greater reduction in physiological arousal (HR and BPSYS), consistent with Dunn et al.’s [36] findings for participants with poor interoceptive ability.

## 4. Data Comparison of Experiments 1 and 2 between the Old Adult Groups

In order to examine whether low interoception influences the relationship between physiological arousal and emotional experience, we compared data from the old adult group in Experiment 1 and the old data group in Experiment 2. No demographic differences were found between the groups (see Table A4 in Appendix B). The two old adult groups showed similar scores in the sub-scales of the BPQSF (see Table A5 in Appendix B). Further, no significant effects were found for the picture type or content in the three physiological indices (HR change, BPDIA change, and BPSYS change). The results suggest that no significant differences were found between the old adult groups in the two experiments.

## 5. General Discussion

Our aim was to examine if there is a decrease in interoception in old age and if this decrease affects emotional experiences. In Experiment 1, young adults demonstrated a significant positive correlation between physiological arousal and self-reported arousal while in older adults no correlation was found. Old participants showed only a negative directional trend between physiological arousal and self-reported arousal, thereby no significant association between their subjective experience of arousal and the objective measurement of their arousal was found (i.e., interoceptive accuracy). As mentioned earlier in this paper, this could reflect a reduction in interoception in old age but can also be a result of intra-individual differences. Older adults also rated their interoceptive awareness as reduced compared to young adults. In Experiment 2, we found significant negative associations between physiological arousal and interoceptive awareness in old age. This means that participants who reported low interoception also showed fewer physiological changes in arousal. This may indicate a normal interoception, as participants reported arousal according to the physiological arousal they experienced (that is, the actual measures of arousal and the self-report matched). However, further observation of the finding shows that these participants showed inconsistent changes in physiological arousal (above and below baseline). In other words, even when they experienced higher arousal, they reported lower interoception. Therefore, we believe this supports Mendes’ idea of connecting blunted physiological arousal to low interoceptive ability in old age [75].

In addition, low interoception (interoceptive awareness) in old age was found to be associated with emotional experience, which focuses less on negative emotional information. These data confirm previous evidence of a decline in interoception in old age (e.g., [84,85,86]) and suggest that changes in interoception do affect emotional experiences in aging (e.g., [75]).

Following our results, we believe that the positive emotional effect of less negative experience seen in old age [57,58,59], as well as the decrease in physiological functioning [40] could be explained by interoception ability and the idea of maturational dualism [75]. According to Mendes [75], the positive effects seen in old age are motivated by the inability to access internal information to determine how one feels, therefore attention is directed to emotional value and external cues from the environment, and in line with the SST [70], toward more positive information. It could also be explained by theories of bodily feedback [2,11,12,106,107]. For example, the conceptual act theory (CAT; cf. [106,107]), suggests that emotions emerge from our perception of the internal state of our bodies, external representations from the present context, attention, prior experience, and knowledge about emotion categories. According to this theory, our research findings could be explained by the knowledge of emotional experiences that older adults possess collected over their lifetimes. Even in the absence of robust physiological arousal and the low perception of it (low interoception), knowledge about the internal and external cues associated with emotions could potentially be used to rise above the physiological decline [107]. It is important to note that even though this positive emotional effect is well established in the literature, it was not examined in our study since our second experiment included only older adults, and no comparison of emotional experience was made to young adults.

Another possible explanation of our findings is through the theory of strength and vulnerability integration (SAVI) [71]. This theory suggests that it may be easier to regulate one’s emotions when one doesn’t experience very arousing intense bodily changes. These explanations, along with our findings, highlight the important role of interoception in the gap between body and mind, in general, and, specifically, in old age.

Comparing findings from Experiments 1 and 2, we found a certain extent of similarities. For example, we noticed that both old groups from the two experiments showed a similar negative directional correlation between physiological and emotional arousal (in Experiment 2) and self-reported physiological arousal (in Experiment 1). This negative correlation was opposite to the significant positive correlation found in young adults. These results replicate previous findings in young adults (e.g., [36]) and support the suggestion that the ability to identify physiological arousal declines in old age.

Regarding our secondary hypotheses, we found that negative pictures were reported as more arousing than positive pictures, as in Experiment 1, and were also reported as less pleasant, replicating previous findings (e.g., [44,62,63]). As opposed to Experiment 1, no HR deceleration trend was found for picture type while an inconsistent general reduction was found in the physiological arousal measures. This may be related to the fact that the majority of old participants used BP medications (70.3%). As shown in Experiment 1, the use of these medications blunted changes in arousal, as can be seen in the smaller changes from baseline during emotional stimulation. The large percentage of BP medication users could also explain why no correlation was found between the BA sub-scale and BP medications in Experiment 2.

The finding that BP medications have a strong mediating effect on interoception ability is surprising and very important. The aging population is at increased risk for hypertension [108,109]; thus, the use of BP medications is quite common. The effect of medication use is not novel, but the parsimonious way it contributes to the explanation of our results is surprising. In our study, BP medication was found explanatory of age differences. This makes conclusions regarding aging processes less clear and may underly some of the discrepancies we see in the literature. It is not clear whether other studies that have examined interoception in old age (e.g., [85]) treated BP medication use as a relevant factor. Since we discovered this effect post hoc, we were unable to thoroughly explore this variable. It is, therefore, important for future studies to refer to BP medication use as a significant factor in examining interoception in old age.

This study has several limitations. First and foremost, there was no correlation between physiological arousal and emotional experience in Experiment 2, so we could not examine if interoception mediated the relationship between the two variables. It is methodologically challenging to create a clear examination of this relation; nevertheless, we presented for the first time a study that examined all variables together.

Second, due to technical limitations of the BP-HR watch, we were not able to measure mid-baseline levels between each set of pictures, but only at the beginning of the experiment. This may have affected the exact level of changes in the physiological parameters. In addition, since the BP-HR watch produced a measurement every 30 s, each set of pictures was presented twice to calculate an average for each subject. It is possible that the habituation effect was developed and affected the level of physiological arousal, thus encouraging a lack of significant differences in the physiological measures.

A third limitation concerns the picture content. Our selection of pictures was limited due to ethical considerations and, therefore, certain contents, known to be particularly arousing (e.g., naked people or sexual relations) were excluded. Perhaps the choice of relatively conservative pictures resulted in a modest effect on the BP index. This is in accord with Charles [71] who predicted that highly arousing situations may result in a prolonged increase in physiological arousal.

Lastly, it would have been more beneficial to examine a larger sample size, taking the large individual differences into consideration. Even though we determined the number of participants by using a G*Power statistical analysis (version 3.1.9: [110]), the large variability affected statistical power and generalization to the entire population. Future research should include considerations of these limitations.

Experimentally, we recommend adding a young comparison group when evaluating emotional experience to allow full age-related exploration of interoception as a mediator and to create an option of comparing the emotional experience between different ages. It is worth noting that many aspects may contribute to interoceptive ability and should be considered in future studies. Here, for example, our participants attended, as part of the communal residency, joint–similar sports programs, had similar living arrangements (e.g., walking from their house on the first floor to a central communal dining room), the same air-conditioning environment throughout the day, and were all without mental health diagnoses.

In conclusion, this work demonstrates that interoception ability declines with age. Moreover, we demonstrate that low interoceptive awareness is accompanied by less negative emotional experiences in old age. We also revealed the important mediating role of BP medications on interoception. These findings highlight the importance of understanding interoception ability changes in old age as it seems to play a role in the body–mind gap, between physiological changes and the emotional experiences of older adults.

## Figures and Tables

**Figure 1 brainsci-12-01398-f001:**
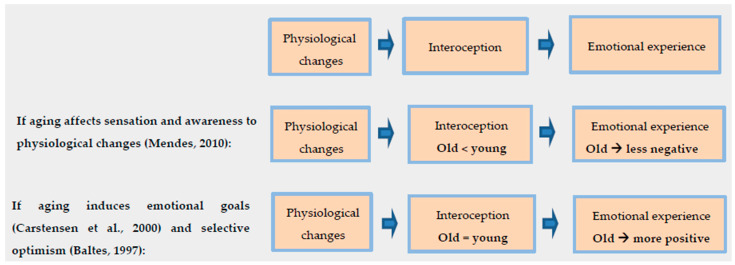
The role of interoception with different predictions following relevant theories [69,73,75].

**Figure 2 brainsci-12-01398-f002:**
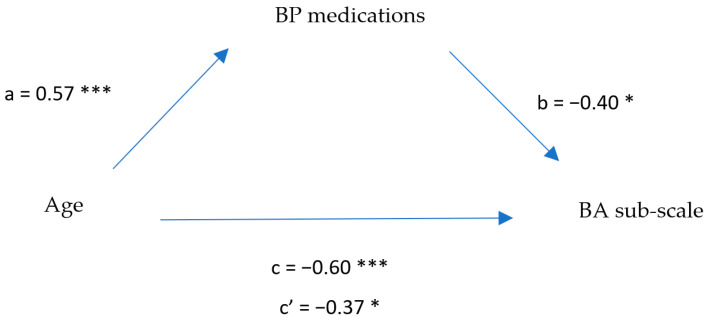
Partial mediation model: BP medications as a mediator in the relationship between age and body awareness (BA) sub-scale. * *p* < 0.05. *** *p* < 0.001. BP medications variable was encoded as dummy variable (0 = no usage, 1 = use of BP medications).

**Table 1 brainsci-12-01398-t001:** Experiment 1—demographic variables.

		General	Young Adults	Older Adults	Parameter	*p*<
		*n* = 37	*n* = 17	*n* = 20		
Age		54.11 (27.68)	25.72 (3.21)	79.65 (5.72)	*t* = −35.27	0.05
Weight		69.5 (16.19)	66.22 (14.5)	72.45 (17.41)	*t* = −1.19	ns
Years of education		13.26 (2.83)	12.83 (1.98)	13.65 (3.42)	*t* = −0.89	ns
Gender (by females)		23	13	10		
Family status						
	Single	15	15			
	Married	14	1	13		
	Divorced	1	1			
	Widow	7		7		
Work status						
	Work	20	14	6		
	Unemployed	11	3	8		
	Volunteer	6		6		
Hours of work or volunteering per week		15.21 (14.25)	21.33 (14.20)	9.70 (12.15)	*t* = 2.72	0.05
BP medications		11		11		
Duration of taking the medication		9.75 (6.76)		9.75 (6.76)		
Psychiatric medication		2	1	1		
Diabetes		1		1		
Current use of cigarettes		9	8	1		
Current alcohol use		1	1	0		
Mean HR change		2.02	2.09 (10.78)	1.96 (9.17)	*t* =0.25	ns
Mean BPDIA change		0.76	0.76 (3.81)	0.76 (3.24)	*t* = −0.01	ns
Mean BPSYS change		1.12	1.22 (5.82)	1.02 (4.97)	*t* = 0.61	ns
Sub-scale body awareness (BA)			82.94 (17.65)	60.00 (15.88)	*t* = 4.16	0.05
Sub-scale supradiaphragmatic reactivity (SUPRAR)			25.24 (5.18)	24.60 (7.19)	0.30	ns
Sub-scale subdiaphragmatic reactivity (SUBR)			12.12 (4.53)	10.25 (3.06)	1.49	

Pearson correlations and significance of demographic variables see Table A2 in Appendix B.

**Table 2 brainsci-12-01398-t002:** Means, standard deviations (SD), statistical parameters, and significance levels of the physiological indices according to the usage of BP medications.

	Use BP Medications (*n* = 11)	No Use BP Medications (*n* = 26)	F _(1,35)_	Effect Size- ηP^2^
	Mean (SD)	Mean (SD)		
HR change	0.07 (6.43)	−4.36 (4.46)	5.81 *	0.14
BPDIA change	−0.14 (2.40)	−1.53 (1.57)	4.40 *	0.11
BPSYS change (normally distributed)	0.09 (3.48)	2.41 (2.42)	6.35 *	0.15

* *p* < 0.05.

**Table 3 brainsci-12-01398-t003:** Pearson correlations between the three physiological indices: mean HR change, mean BPDIA change, mean BPSYS change, and self-reported physiological arousal for the two age groups.

	Young Adults	Older Adults
	1. Mean self-report	1. Mean self-report
2. Mean HR change	0.51 *	−0.20
3. Mean BPDIA change	0.61 **	−0.14
4. Mean BPSYS change	0.53 *	−0.18

* *p* < 0.05. ** *p* < 0.01.

**Table 4 brainsci-12-01398-t004:** Pearson correlations and significance of the variables of Experiment 1.

Indices	1	2	3	4	5	6
1. BA sub-scale	-					
2. Age	−0.60 *	-				
3. Mean self-report	−0.19	0.30	-			
4. Mean HR change	0.27	−0.25	−0.05	-		
5. Mean BPDIA change	0.21	−0.23	−0.12	0.99 **	-	
6. Mean BPSYS change	0.26	−0.26	−0.07	0.99 **	0.98 **	-

* *p* < 0.05. ** *p* < 0.01.

**Table 5 brainsci-12-01398-t005:** Pearson correlations (two-tailed) between age, BP medications, and BA sub-scale.

	1.	2.	3.
1. Age	-		
2. BP medications	0.57 **	-	
3. BA sub-scale	−0.60 **	−0.62 **	-

** *p* < 0.01.

**Table 6 brainsci-12-01398-t006:** Means (M) and standard deviations (SD) for the subjective ratings of picture type and picture content for each age group. * Represents a significant difference between age group and Picture type.

	Picture Content	General (*n* = 37)	Young Participants (*n* = 17)	Old Participants (*n* = 20)
**negative**		3.91 (2.03)	3.20 (1.40)	4.51 (2.30)
	Domestic violence *	4.55 (2.26)	3.67 (1.47)	5.30 (2.57)
	Environmental pollution *	3.04 (2.29)	2.08 (1.43)	3.85 (2.60)
	Injured people	4.97 (2.41)	4.41 (1.91)	5.45 (2.70)
	Injured animals	4.50 (2.27)	4.08 (1.68)	4.85 (2.66)
	Negative emotion	3.64 (2.15)	3.05 (1.69)	4.15 (2.41)
	Negative situations *	2.91 (1.81)	2.29 (1.62)	3.45 (1.82)
	Old people *	3.75 (2.24)	2.82 (1.50)	4.55 (2.49)
**positive**		2.78 (1.78)	2.09 (1.45)	3.36 (1.87)
	Babies *	3.36 (2.45)	2.31 (1.87)	4.25 (2.57)
	Couples in love *	3.12 (2.19)	2.34 (1.83)	3.79 (2.29)
	Desserts	2.17 (1.67)	2.37 (1.98)	2.00 (1.40)
	Happy children **	3.17 (2.53)	1.79 (1.37)	4.33 (2.73)
	Pets *	2.47 (1.91)	1.74 (1.51)	3.10 (2.02)
	Sport	2.15 (1.61)	1.84 (1.73)	2.42 (1.50)
	views	3.06 (2.23)	2.34 (1.86)	3.67 (2.38)

* *p* < 0.05. ** *p* < 0.01.

**Table 7 brainsci-12-01398-t007:** Experience 2—demographic variables.

		General
		*n* = 27
Age		77.81 (5.81)
Weight		73.37 (12.92)
Years of education		11.41 (5.9)
Gender (by females)		15
Family status		
	Single	1
	Married	16
	Divorce	1
	Widow	9
Work status		
	Work	9
	Unemployed	13
	Volunteer	5
Hours of work or volunteering per week		9.72 (14.42)
BP medications		19
Duration of taking the medication		12.28 (15.01)
Psychiatric medication		1
Diabetes		4
Current use of cigarettes		2
Current alcohol use		1
Mean HR change		−0.07 (12.13)
Mean BPDIA change		−0.13 (4.23)
Mean BPSYS change		−0.09 (6.56)
Sub-scale body awareness (BA)		55.78 (17.16)
Sub-scale supradiaphragmatic reactivity (SUPRAR)		20.74 (6.55)
Sub-scale subdiaphragmatic reactivity (SUBR)		9.85 (3.57)

**Table 8 brainsci-12-01398-t008:** Means, standard deviations (SD), statistical parameters, and significance levels of the BPQSF subjective rating by gender.

	Females *n* = 15	Males *n* = 12	*t*(25)	95% CI
Sub-scale body awareness (BA)	63.13 (17.53)	46.58 (11.81)	−2.80 **	[−28.74, −4.36]
Sub-scale Supradiaphragmatic reactivity (SUPRAR)	24.33 (6.79)	16.25 (1.76)	−4.00 ***	[−12.24, −3.92]
Sub-scale Subdiaphragmatic reactivity (SUBR)	11.67 (3.64)	7.58 (1.78)	−3.55 **	[−6.45, −1.72]

** *p* < 0.01. ****p* < 0.001.

**Table 9 brainsci-12-01398-t009:** Pearson correlations and significance of Experiment 2 variables.

Indices	1	2	3	4	5	6	7	8
1. BA	-							
2. Mean change in HR	−0.37	-						
3. Mean change in BPDIA	−0.38 *	0.96 **	-					
4. Mean change in BPSIS	−0.38 *	0.96 **	0.97 **	-				
5. EE-negative	−0.38 *	0.11	0.12	0.15	-			
6. EE-positive	−0.02	0.21	0.27	0.19	−0.11	-		
7. EA-negative	0.34	−0.30	−0.34	−0.29	−0.51 **	0.00	-	
8. EA-positive	−0.04	−0.39 *	−0.32	−0.41 *	0.06	0.03	−0.10	-

* *p* < 0.05. ** *p* < 0.01.

**Table 10 brainsci-12-01398-t010:** Means and standard deviations (SD) of the subjective ratings of emotional experiences and emotional arousal in response to picture type and picture content (N = 27).

Picture Type	Picture Contents	Emotional Experience	Emotional Arousal
negative		1.07 (1.30)	6.00 (1.04)
	Domestic violence	0.56 (1.19)	7.07 (1.21)
	Environmental pollution	0.59 (1.25)	6.33 (1.57)
	Injured people	0.52 (1.19)	6.89 (1.50)
	Injured animals	0.74 (1.48)	6.19 (1.52)
	Negative emotion	1.93 (1.57)	5.07 (1.44)
	Negative situations	2.26 (1.53)	5.19 (1.08)
	Old people	1.93 (2.06)	5.11 (1.48)
positive		6.56 (.85)	4.33 (1.78)
	Babies	7.11 (1.37)	4.19 (2.56)
	Couples in love	7.26 (1.09)	5.19 (2.42)
	Desserts	6.07 (1.61)	4.93 (2.09)
	Happy children	6.78 (1.53)	4.56 (2.33)
	Pets	6.11 (1.45)	3.81 (1.42)
	Sport	4.78 (1.31)	3.93 (1.75)
	views	7.33 (1.24)	3.41 (2.99)

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
