# Peer review of "Interoception in Old Age"

_brainsci, 2022, doi:10.3390/brainsci12101398_

Round 1

Reviewer 1 Report

This manuscript investigates the association between aging, interoceptive awareness, physiological arousal, and emotional arousal. They conducted two experiments (Experiment 1 with younger and older adults, and Experiment 2 with older adults only). The purpose of Experiment 1 was to examine the aging effects on interoception and the purpose of Experiment 2 was to further investigate the association between interoception and emotional experience in older adults.

They have five main findings. First, the association between self-reported interoceptive arousal and physiological arousal was not significant in older adults while those were positively correlated in younger adults (Experiment 1), which indicates the aging effects on the association between those (mediating effect of aging is not clear since they did not conduct any mediation analyses here). Second, they found that older adults reported lower scores of body awareness (Experiment 1), which additionally supports the notion that aging reduces interoceptive arousal/awareness. Third, higher emotional arousal was associated with lower HR and BPSYS in older adults regardless of the emotional valence (Experiment 2). This third finding was not clearly hypothesized in the Introduction. Fourth, lower scores of body awareness were associated with more pleasant emotional experiences overall during the emotional stimuli task, and lower scores of body awareness were associated with less negative emotional experiences in the face of negative stimuli in older adults (Experiment 2). Note that this fourth finding was described in a wrong way if I understand the study design correctly (e.g., L696) since body awareness was trait-like scales measured at the baseline before the emotional stimuli task while the pleasantness was state-like self-report measured during the stimuli task. Fifth, lower scores of body awareness were associated with higher (or greater changes?? It was not clear) physiological arousal (BPDIA and BPSYS) in physiological arousal (Experiment 2). Those associations were negative correlations; however, the authors interpreted them as: “This means that reports of low body awareness corresponded with smaller changes in physiological arousal from baseline, suggesting that low interoception was associated with fewer changes in physiological arousal” (L707-709), and this interpretation does not look appropriate (direction is opposite).

Overall, this study offers insight into the role of aging and interoception in emotional experiences. The study setup is somewhat unclear and there are also some important omissions with respect to the reported results and some aspects need further discussion. The introduction could be more concise. Many variables were used without clear definitions (e.g., emotional arousal, emotional pleasantness, interoceptive ability, etc.).

Major points:

1.    As I stated above, the third finding was not clearly hypothesized in the Introduction.

Hypotheses should be more clearly stated, for example: (a)The changes in self-reported physiological arousal at (time point) relative to (time point) and the changes in physiological arousal (i.e., BP and HR) at (time point) relative to (time point) would be positively associated in the young adults' group, but not in the old adults' group (in line with [36]); and b) older adults would report lower levels of body awareness measured by BA subscale than young adults (in line with [84,86]). In older adults, (c) a lower level of interoception (what? ability, awareness?) at baseline would be associated with greater levels of pleasant experiences during the emotional stimuli task (in which condition? Positive or negative? Or overall?), and with (d) fewer changes in physiological arousal measured by (what? BP and HR?) during the emotional stimuli task (in which condition? Positive or negative? Or overall?).

2.    As I summarized above, the fourth finding was described in a wrong way if I understand the study design correctly (e.g., L696) since body awareness was trait-like scales measured at the baseline before the emotional stimuli task while the pleasantness was state-like self-report measured during the stimuli task.

3.    Similarly, the fifth finding was not clearly stated. Lower scores of body awareness were associated with higher (or greater changes?? It was not clear) physiological arousal (BPDIA and BPSYS) in physiological arousal (Experiment 2). Those associations were negative correlations; however, the authors interpreted them as: “This means that reports of low body awareness corresponded with smaller changes in physiological arousal from baseline, suggesting that low interoception was associated with fewer changes in physiological arousal” (L707-709), and this interpretation does not look appropriate (direction is opposite). I might have misunderstood something here, but I appreciate any clarifications.

4.    Other than the above five findings should be less emphasized since those are not related to a priori hypotheses including BP medications.

5.    Abstract should be clearly stated with hypotheses and findings (the current Abstract does not accurately reflect findings but rather mixed with expectations).

6.    L 52. The authors mentioned that the association between interoception (awareness and accuracy) and emotional experience is well established in studies; however, it appears that higher interoception is not only associated with negative emotions [27], alexithymia [34], and higher levels of emotional arousal [28,29,35-37], but also is associated with better self-regulation capacities [38] and better emotion regulation in response to negative affect. Therefore, the link between prior studies and hypothesis (c) is weak.

7.    L183 and Figure1. The authors hypothesized the mediating role of interoception between physiological changes and emotional experiences; however, hypotheses (c) and (d) are formulated as interoceptive awareness influences both emotional experiences and physiological changes, which does not indicate the mediational role of interoceptive awareness. I do not see the point of Figure 1 in this manuscript.

8.    Figure 1. In the third model, interoceptive awareness mediates the relationship between physiological changes and emotional experiences, however, there would be no aging effects although it says that the older populations would experience more positive emotions than younger populations. This logical flow is not clear in the introduction. How was the third model formulated? Which model was most likely appropriate based on the current results or none of them?

9.    Can the authors clarify the difference between “interoceptive ability” and “interoceptive awareness/accuracy”?

10. L267. “Interoception ability was measured by the rating of perception of internal physiological activity or changes during the emotional task.” Can the authors clarify the definition of “interoceptive ability” and how did they measure this concept?  How did they use the former instruction (physiological activity) and the latter instruction (changes during the emotional task)? Was this asking the interoceptive awareness “right now” or changes in interoceptive awareness from the last time point to the current time point? Similarly, when they mentioned this interception ability, it was not clear if they indicated the ability at the baseline or changes in ability during the task (especially in Experiment 2).

11. L323. BPDIA and mean BPSYS needs to be explained for the first use of the abbreviations.

12. I assume that BP medication users were all in the older group. There is a chance that age and medication use confound each other in regards to BP variables. I suggest breaking down Table 1 into two groups (BP medication use vs no use in the older group and younger group) to examine BP medication use influenced BP variables in the older group.

13. How was the distribution of BP variables? Were those log-transformed for the statistical analyses or were those normally distributed without any transformations?

14. L330. Table 3 should be table 2?

15. L365. The older group reported higher arousal compared to young adults. It is not clear whether this statement is related to the combined picture ratings (negative+positive) or within each picture type. I would suggest analyzing and reporting separately to answer model 2 (expecting older adults would have lower arousal levels in response to negative stimuli) and model 3 (expecting older adults would have higher arousal levels in response to positive stimuli) in Figure 1.

16. L453. Incomplete sentence.

17. Table 5. The correlation between age and BP medications is negative, indicating that older adults use less BP medications. Is this correct? Same as the relationship between BP medications and BA sub-scale, which is a negative association, indicating less BP medication use was associated with higher scores on BA sub-scale. Please also check Figure 2.

18. L530. “This finding may support a blunting effect of BP medication use on internal physiological processes and on interoception ability.” Why did the authors interpret the result as a blunting effect of BP medication since BP medication users had a higher BA subscale?

19. None of the hypotheses were formulated to test the mediating role of interoceptive awareness.

20. It would be helpful for readers if the authors could organize the results section as follows: first, describe demographic variables, and then explain the results related to the main hypotheses, followed by exploratory analyses.

21. The authors mentioned that “changes in physiological arousal” several times in the manuscript; however, it was not clear how they calculated “changes” (e.g., relative to which time point or different conditions, etc.). Similarly, do the associations between interoception and changes in physiological arousal indicate those of which between changes in state-interoception awareness or trait-like measurement of interoception awareness at baseline and changes in physiological arousal?

22. L712. “In order to examine whether low interoception influences the relationship between physiological arousal and emotional experience, we compared data from the old adult group in Experiment 1 and the old data group in Experiment 2.” I do not think this sentence is relevant and needs to be rephrased to make it clear what is the purpose of comparing the data from Experiments 1 and 2.

23. L732. “This may indicate a normal interoception, as participants reported arousal according to the physiological arousal they experienced.” I am not sure what the authors wanted to say here. What is “normal interoception”?

24. L 739. The authors claimed that reduced interoception in older age did affect less focus on negative emotional experiences. However, it could be that aging influenced less focus on negative emotional experiences which might have contributed to the lower interoceptive awareness.

25. Does the author have data of history and doses of BP medication?

26. Experiment 2. All the correlation analyses related to hypothesized variables should be reported with a table (not only significant results).

27. L742. “Following our results, we believe that the positive emotional effects seen in old age [57-59], as well as the decrease in physiological functioning [40] could be explained by interoception ability and the idea of Maturational Dualism [75].” I do not see any results related to positive emotions (response to positive stimuli?) and interoception in Experiment 2.

28. Since aging affects average HR and BP in general (regardless of emotional experiences), how did the authors account for this general aging effect on physiological measures? A lack of neutral pictures is one of the limitations.

29. Tables should be formatted appropriately instead of listing the SPSS results as tables.

30. The motivation of BP medication mediation analysis should be clearly stated in the Introduction, or at least it should be clearly stated as exploratory analyses. Also, since they could not replicate the BP medication effect in Experiment 2, the emphasis on mediating effects of BP medication should be toned down. I assume that aging and the use of BP medication are confounding.

Minor points:

Some citations are missing.

Careful grammar and spell check will be required.

Reviewer 2 Report

The manuscript „ Interoception In Old Age “ examines interoceptive abilities and their influence on emotion regulation in old participants. The manuscript contains two different interesting experiments. Even though, the manuscript is is generally well written and investigates an important topic, a couple of issues dampened my enthusiasm for the manuscript, which I will address in the following.

Abstract

There is information on the sample missing, how many subjects participated?

Introduction

-          In line 30 there is something wrong with the reference it says “REF OF FELDMANN-BARRET”

-          There has been an ongoing debate on the facets of interoception and terms are frequently misused. There are actually three facets of interoception: interoceptive sensibility is the subjective estimation of one’s own interoception. Here the authors refer to this facet as interoceptive awareness. But interoceptive awareness is the metacognitive assumption about one’s own interoceptive accuracy. A good example is the MAIA questionnaire, which measures sensibility, and the Heart Beat Perception Task from Schandry, which is also mentioned by the authors, which measures both interoceptive accuracy and awareness over specific formula. The authors need to differentiate correctly between the facets. The terms interoceptive sensibility and awareness are mixed uo in many studies and this is a problem in research on interoception because it leads to wrong conclusions.

Methods

-          Could the authors explain why they did not stick with common measures for interoceptive accuracy like the Heartbeat Perception Task or the Heartbeat Detection Task but rather measured BP and HR and subjective self-report?

-          Which watch did the authors use for measuring BP and HR? There are huge differences in accuracy depending on the brand. It just says “FITNESS TRACKER” in the parentheses.

-          I cannot quite follow the description on reliability measures for the BPQSF but I saw that the authors provided internal consistency for the present sample as well in the result section, maybe this could already be mentioned in the measure section

-          Could the authors provide a power calculation for their analyses since the sample seems rather small. They mention in the discussion that they determined the sample size with g*power but what were the used parameter for this?

Results

-          The font size changes in the beginning of the result section line 315-319

-          The authors provide evidence for gender differences and also for correlations between interoception and weight and medication, why did the authors not control for these variables in their regression analyses? Interestingly, a mediation was calculated afterwards.

 Continuing with the second experiment, the font size is very small in comparison to the rest of the manuscript.

Discussion

-          The authors state that it is surprising that BP medication has an influence on interoception. I, however, am surprised from this statement. It has been well known that interoceptive abilities are influenced by multiple factors such as weight, gender, medication. These three aspects have been mentioned by the authors but there are several other important aspects that have not been mentioned and it does not become clear whether this was even assessed in participants. Mental disorders have been associated with interoceptive deficits multiple times as well as endurance sport.

Round 2

Reviewer 1 Report

Thank you so much for all the responses to my comments. Although I see the authors made a lot of effort to improve their manuscript, unfortunately, I do not see significant improvements as I expected. I hope that my comments below will help the authors improve the quality of the manuscript.

1. Regarding point 1. Hypothesis.

I appreciate the authors efforts to make the hypothesis clear.

“during the task”

Since the authors did not explain what task they used, I suggest they briefly explain the task before they introduced the hypothesis. Unless the readers would not understand what is “after 30 seconds and then every 30 seconds” in this hypothesis (a).  

2. Now I realized that changes in self-reported physiological arousal and change in BP and HR were measured every 30 seconds during the task. I read “Measures” “Emotional Stimuli.” How long the task was? How did the authors calculate the correlation using the values measured every 30 secs? This should be clearly stated in the method section.

3. Regarding point 12. Table 1.

What I suggested was to reconstruct Table 1 to show the basic information between the two groups you are interested in.

Young adults            Old adults

(n=17)                         (N=20)

Age

Gender

Weight

Smoking

Daily alcohol consumption

Use of BP medication

Use of psychiatric medication

Mean change in HR

Mean change in BPDIA

Mean change in BPSYS

Baseline (?) BPQSF

BA subscale

            SUPRAR subscale

            SUBR subscale

I see that the authors summarized that information as the Appendix table and BP medication use was all in the old adults group. Then the authors should compare the Use of BP medication and the No Use of BP medication in the old adults only since no one in the young adults group did use BP medication to say BP medication influenced interoception.

4. Regarding points 17 and 18. Table 6 (previously Table 5). I would suggest presenting this table using a code use of BP medication as greater than no use of BP medication to be consistent with Table 1 and other sentences in the main text. Just recode the categorical value as 0 = no use and 1 = use, or 1 vs.2, etc. It is easy for readers to understand the relationship.

5. Regarding point 20.   The authors said that they agreed to restructure the results section but I did not see the change in the main text.

6. Regarding point 21. I did not understand their response and did not see the edits in the main text.

7. Regarding point 23. I still do not understand what “the actually measured arousal and the self-report matched’ means? How did the authors identify “matched” from the data and results?

8. Regarding point 24. I simply pointed out that there would be a bi-directional relationship between reduced interoception and less focus on negative emotional experiences by aging. However, the authors interpreted it as one direction which should be corrected in the main text.

9. Regarding point 25. Where? I meant adding how long they were taking BP medications and the dose of BP medications. I do not see it in any tables.

10. Regarding point 27. Okay, and what changes did the authors make to respond to my comment? Since they did not investigate anything related to positive emotions, this sentence is not relevant.
